# From Localized Mild Hyperthermia to Improved Tumor Oxygenation: Physiological Mechanisms Critically Involved in Oncologic Thermo-Radio-Immunotherapy

**DOI:** 10.3390/cancers15051394

**Published:** 2023-02-22

**Authors:** Peter Vaupel, Helmut Piazena, Markus Notter, Andreas R. Thomsen, Anca-L. Grosu, Felix Scholkmann, Alan Graham Pockley, Gabriele Multhoff

**Affiliations:** 1Department of Radiation Oncology, University Medical Center, University of Freiburg, 79106 Freiburg, Germany; 2German Cancer Consortium (DKTK), Partner Site Freiburg, German Cancer Research Center (DKFZ), 69120 Heidelberg, Germany; 3Department of Anaesthesiology and Intensive Care Medicine, Charité-University Medicine, Cooperative Member of Freie Universität Berlin and Humboldt Universität zu Berlin, 10117 Berlin, Germany; 4Department of Radiation Oncology, Lindenhofspital Bern, 3012 Bern, Switzerland; 5Biomedical Optics Research Laboratory, Department of Neonatology, University Hospital Zurich, University of Zurich, 8091 Zurich, Switzerland; 6Institute of Complementary and Integrative Medicine, University of Bern, 3012 Bern, Switzerland; 7John van Geest Cancer Research Centre, Nottingham Trent University, Nottingham NG11 8NS, UK; 8TranslaTUM—Central Institute for Translational Cancer Research, Technische Universität München (TUM), 81675 Munich, Germany; 9Department of Radiation Oncology, Klinikum Rechts der Isar (TUM), 81675 Munich, Germany

**Keywords:** radio-oncology, mild hyperthermia, immuno-oncology, enhanced tumor blood flow, improved tumor oxygenation, physiological responses, transient changes, sustained effects, pleiotropic hyperthermia effects

## Abstract

**Simple Summary:**

Mild hyperthermia (mHT, 39–42 °C) is a potent modality when combined with existing radio-, chemo-, or immunotherapy, leading to enhanced microcirculatory blood flow and improved tumor oxygenation. Currently, the mechanisms responsible for these mHT-related effects are not fully understood, and the extent and kinetics of therapy-promoting changes are not yet completely clarified. Herein, we review and discuss mHT-induced biological mechanisms that are relevant to radio-oncology and immunotherapy. The short-term increase in tumor perfusion induced by mHT may be caused by vasodilation of co-opted vessels and upstream normal tissue vasculature, as well as decreases in viscous resistance to flow. More sustained effects seem to result from a reduction in interstitial fluid pressure and VEGF-triggered angiogenesis. Increased microcirculatory tumor blood flow after mHT seems to be the prime driver for the enhanced tumor oxygenation, and this is supported by increasing O_2_ diffusivities and O_2_ extraction and facilitated O_2_ unloading from oxyhemoglobin due to right-shifts of the HbO_2_-dissociation curve by hyperthermia per se and intensified tumor acidosis.

**Abstract:**

(1) Background: Mild hyperthermia (mHT, 39–42 °C) is a potent cancer treatment modality when delivered in conjunction with radiotherapy. mHT triggers a series of therapeutically relevant biological mechanisms, e.g., it can act as a radiosensitizer by improving tumor oxygenation, the latter generally believed to be the commensurate result of increased blood flow, and it can positively modulate protective anticancer immune responses. However, the extent and kinetics of tumor blood flow (TBF) changes and tumor oxygenation are variable during and after the application of mHT. The interpretation of these spatiotemporal heterogeneities is currently not yet fully clarified. (2) Aim and methods: We have undertaken a systematic literature review and herein provide a comprehensive insight into the potential impact of mHT on the clinical benefits of therapeutic modalities such as radio- and immuno-therapy. (3) Results: mHT-induced increases in TBF are multifactorial and differ both spatially and with time. In the *short term*, changes are preferentially caused by vasodilation of co-opted vessels and of upstream normal tissue vessels as well as by improved hemorheology. *Sustained* TBF increases are thought to result from a drastic reduction of interstitial pressure, thus restoring adequate perfusion pressures and/or HIF-1α- and VEGF-mediated activation of angiogenesis. The enhanced oxygenation is not only the result of mHT-increased TBF and, thus, oxygen availability but also of heat-induced higher O_2_ diffusivities, acidosis- and heat-related enhanced O_2_ unloading from red blood cells. (4) Conclusions: Enhancement of tumor oxygenation achieved by mHT cannot be fully explained by TBF changes alone. Instead, a series of additional, complexly linked physiological mechanisms are crucial for enhancing tumor oxygenation, almost doubling the initial O_2_ tensions in tumors.

## 1. Introduction

Physiological responses of malignant tumors upon mild hyperthermia (mHT, i.e., therapeutically heating malignant tumors at a (preferred) temperature range of 39–42 °C for 30–60 min [1,2,3,4,5]) have been reviewed during the last two decades with different focuses and perspectives (e.g., [6,7,8,9,10,11,12,13]). Preclinical hyperthermia (HT) studies published since the late 1970s and into the early 1980s are unfortunately limited in terms of their relevance for the clinical setting, e.g., due to the (i) use of impractical heating techniques, (ii) tissue temperature levels >43 °C (combined with long exposure times) which are cytotoxic per se and not achievable in clinical settings, (iii) bulky, fast-growing rodent tumors exhibiting very low blood flow rates do not reflect the clinical situation, (iv) incorrect and misguided conclusions from *in vitro* experiments, and (v) incorrect and misguided interpretation of biological responses upon mHT [14].

As a multifaceted adjuvant cancer treatment modality, mHT triggers a series of therapeutically relevant biological mechanisms at this temperature level without direct cytotoxicity. Localized mHT is usually administered as an adjuvant modality to primarily improve the therapeutic efficacy of “concurrent” radio- and/or chemotherapy and in heavily pretreated patients with recurrent cancers [15,16]. Delivered within a close time frame before external radiation (RT) and/or chemotherapy (CT), mHT has unequivocally shown distinct beneficial effects in numerous clinical studies (e.g., [17,18,19]).

mHT is a proven potent radio- and chemosensitizer exerting pleiotropic effects in solid tumors. mHT can be considered as a “general sensitizer” to therapeutic modalities and is a strong modulator of the anticancer immune system:***Hyperthermic radio-sensitization*:** This condition is generally believed to be the result of a “physiological vasodilation”, which increases tumor blood flow (“reperfusion”), and oxygen levels (“reoxygenation”), occasionally lasting up to 24–48 h post-HT [12].***Hyperthermia enhances cytotoxicity of anticancer drugs*:** Besides direct sensitization to a series of anticancer agents (e.g., Cisplatin, Carboplatin, Oxaliplatin, Bleomycin, Doxorubicin), HT can improve the blood-borne delivery (via an increase in heat-induced tumor perfusion and/or a homogenization of blood flow), and enhanced extravasation in the leaky microvasculature of malignant tumors or a temperature-triggered drug release from thermo-sensitive liposomes for localized thermo-chemotherapy.***Hyperthermia inhibits DNA repair enzymes*:** In the upper range of mHT (41–43 °C), several DNA damage repair enzymes responsible for the repair of potentially lethal or sublethal DNA damage have been reported, thus increasing the efficacy of radiotherapy and some chemotherapeutic drugs [1,3,13].***Hyperthermia affects radio-immuno-oncology*:** It is known that hypoxia compromises anticancer immune responses such as reducing the survival, cytolytic, and migratory activity of key effector cells such as natural killer (NK) cells and NK-like T cells, as well as CD4^+^ helper and CD8^+^ cytotoxic T cells, reduces the production of essential “effector” cytokines, as well as fostering an immunosuppressive environment by supporting immunoregulatory Treg cells, myeloid-derived suppressor cells (MDSCs) and inducing the expression of immune checkpoint inhibitors (reviewed in [20]). HT-induced improvements of tumor oxygenation status (“reversal of tumor hypoxia”) and the increased perfusion triggered by mild HT enhances the trafficking of immune cells, and intra-tumoral access to crucial immune regulators such as antibodies and cytokines, all of which are needed to generate effective antitumor immune responses. Hyperthermia is also known to be an effective immune modulator that has multiple effects on the innate and adaptive immune systems (reviewed in [21]).***Hyperthermia and the innate immune system*:** With respect to the *innate immune system*, hyperthermia increases the expression of activation receptors such as NKG2D and MHC class I-related chain A (MICA) on the surface of natural killer (NK) cells, thereby enhancing their antitumor potential [21]. This is confirmed by findings that NK cells are important mediators of antitumor immunity after radiotherapy and hyperthermia [22] and that cells of the innate immune system in patients recover faster when hyperthermia and radio-chemotherapy are combined [23].***Hyperthermia and the adaptive immune system*:** With regards to the *adaptive immune system*, hyperthermia influences all aspects of adaptive antitumor immunity, from the function and antigen presentation capacity of antigen-presenting cells (APCs) to the responsiveness of CD4^+^ and CD8^+^ T-cell populations [23]. Combining hyperthermia with radiotherapy promotes the infiltration of dendritic cells—crucial antigen-presenting cells and initiators of adaptive immune cells—into solid tumors [23], as well as inducing the maturation of DCs and the release of pro-inflammatory cytokines from DCs and macrophages [24,25]. In addition to direct effects on cellular immunity, combining hyperthermia and radiotherapy has also been shown to mediate immune effects via multiple mechanisms, including the release of Danger Associated Molecular Pattern (DAMP) signals such as heat shock proteins (HSPs) and HMGB1 [24,25].

Taken together, these findings and observations demonstrate and support the concept that combining hyperthermia, radiotherapy and immunotherapy with next-generation cellular and antibody-based approaches such as checkpoint inhibition has great therapeutic potential. However, before this approach can be fully exploited, it is crucial to fully understand the physiological consequences of hyperthermia on the tumor microenvironment and identify the optimal doses for achieving the optimal therapeutic conditions.

Based on experimental preclinical and clinical data, it is often concluded that heat-induced increases in TBF are the principal (or even exclusive) drivers of concurrent (“secondary”) changes in the oxygenation status [26] and are principally involved in most of the heat-induced sensitizing mechanisms described above. However, when analyzing the heat-induced changes of tumor blood flow in more detail, it is evident that there are some enigmas (or even deficits) in the interpretation of the results, especially with respect to (i) the genesis of HT-induced flow increases, and (ii) the interpretation of “exclusively flow-related enhancements” of tumor oxygenation status. A more detailed analysis of the functional background clearly shows that mHT-induced secondary effects cannot be fully explained by blood flow improvements alone. Instead, these effects are the result of complex interactions of a multitude of processes. Furthermore, partly dependent on hyperthermia levels and durations, inconsistent directions, as well as highly variable individual extents and kinetics of blood flow changes upon mHT, have also to be considered. In addition, taking into account the pathogenesis-related classification and different timeframes of tumor hypoxia [27], it is rather unlikely that enhancements of the oxygenation status upon mHT closely reflect the improvements in TBF.

## 2. Methods, Search Strategies and Sources of Information

A systematic literature search of relevant research articles and reviews published between 1 January 1980 and 15 November 2022 provided an updated, comprehensive data review (PubMed, Web of Science). Values presented in this condensed article are averaged means. Search items were as follows: tumor hyperthermia, tumor blood flow, tumor perfusion, tumor oxygenation status, tumor oxygen supply, physiological mechanisms, oxygen-dependent radio-sensitization, HT-dependent radio-sensitization, HT-induced immune modulation (and combinations of these terms). This review is also based on one author’s expertise in this research area since the late 1970s (PV).

Exclusion criteria were as follows: (i) preclinical data obtained in fast-growing, bulky rodent tumors (>1% of body weight!), (ii) heating above 43 °C for more than 60 min because information obtained is clinically not translatable or can only be translated to a limited extent, and (iii) immersion of tumors into heated water baths, because the latter can be accompanied by severe edema formation due to osmotic water shifts upon treatment, thus probably concealing HT-related effects [9]. This osmotic water shift adds to the HT-induced edema, thus further expanding the extracellular space [28]. In addition, the whole leg is at elevated temperatures, a condition which might impact blood flow and the temperature distribution within the tumor to be treated (i.e., through “steal phenomena”).

## 3. Results: Assessment of Reliable Experimental and Clinical Data

### 3.1. Transiently Improved Tumor Blood Flow upon Localized Mild Hyperthermia: Potential Mechanisms Involved

#### 3.1.1. Tumor Vascularization and Blood Flow Are Decisive Parameters Critically Affecting Efficacy of Localized Hyperthermia

The vascular physiology of tumors is uniquely different from that of the corresponding normal tissues [29]. When considering the continuous and indiscriminate formation of a vascular network in a growing tumor, five different pathophysiological mechanisms have to be taken into consideration: (i) *co-option (accessing or “hijacking”) of (pre-)existing vessels* at the site of tumor growth, (ii) *angiogenesis* by endothelial sprouting from existing venules, (iii) *vasculo-genesis*, i.e., de novo vessel formation through the incorporation of circulating endothelial precursor cells), (iv) *intussusception* (i.e., splitting of the lumen of an existing vessel into two), and (v) *formation of pseudo-vascular channels*, lined by tumor cells rather than endothelial cells (“vascular mimicry”) [29].

The tumor (micro-)vasculature is characterized by vigorous proliferation, which leads to immature, structurally defective, and in terms of perfusion, ineffective microcirculation [29]. This occurs partly due to missing functional innervation and a lack of pharmacological receptors of contractile elements (smooth muscle cells, pericytes). Relevant thermoregulatory vascular responsiveness and flow regulation within malignant tumors can, therefore, only take place in/via co-opted vessels.

Blood flow rates in malignant tumors vary considerably, ranging from 0.01 to 1.0 mL g^−1^ min^−1^. In squamous cell carcinomas of the head and neck, as well as in colorectal cancers, flow rates up to ≈ 2 mL g^−1^ min^−1^ have been found, whereas, in prostate cancers, flow rates even up to 3 mL g^−1^ min^−1^ have been described [4,29,30]. In addition, variability of TBF flow *within* an individual tumor (i.e., within different tumor sub-volumes, “intra-tumor heterogeneity”) can be in the same flow range as described above for different tumor histologies, stages, and growth sites (“inter-tumor heterogeneity”) [31].

#### 3.1.2. Prime Role of Tumor Blood Flow in Hyperthermia Treatments

Considering the major role of TBF in HT treatments, since the early 1980s, it has mistakenly been postulated that malignant tumors generally have lower blood flow rates than the adjacent normal tissue, thus, favoring quasi-selective heating of solid malignancies due to (i) compromised heat dissipation via convective heat transfer and (ii) the ability for “selective” heat deposition. Using electromagnetic irradiation, this latter notion completely disregards the role of tumor hyperhydration responsible for higher specific heat capacities (“heat storage capacities”) and thermal conductivities [4].

Besides its seminal impact on the heating properties of tumors, the efficacy of TBF can greatly impact parameters of the tumor microenvironment (TME), e.g., the tumor hypoxia and depletion of the bioenergetic status as well as tumor acidity (pH-distribution), i.e., parameters that substantially enhance the cytotoxic effect of HT at 43–45 °C.

As shown in Figure 1A–E, which summarizes own preclinical data published since 1980 [32,33,34,35] (using various heating techniques and detection methods, subtle control of key parameters of the cardiovascular and respiratory system, and of body core temperature), temperature-controlled, local mHT at 39.5–40.5 °C for 30–60 min, regardless of the heating technique (immersion into heated water not included), leads, on average, to transient (during heating and shortly after terminating mHT) and significant increases in TBF with concurrent significant improvements in the oxygenation status during the heating period (Figure 1A). Usually, both enhancements are no longer evident 1 h after heating [32]. At tumor tissue temperatures ≥43 °C for 60 min, a shutdown of TBF is often observed upon heating due to a wide range of mechanisms, including a series of intravascular events, vessel wall alterations, extravascular events, and opposing host tissue/tumor mechanisms (extensively summarized in [33,34]) (Figure 1B,C).

Using water-filtered infrared-A- (wIRA-)heating for 60 min, mild hyperthermia significantly increased TBF by 30–80% above baseline throughout the observation period, with elevations remaining until the end of the treatment period of the tumors (with wet weights ≤ 0.5% of body weight). Upon mHT treatment, *pO*_2_ values increased by 50% on average. These favorable conditions were no longer evident ≈ 1 h post-heat [9,35,36]. HT-induced changes of TBF are often equivocal in that they can change in non-predictable directions and to different extents and durations. In addition, localized divergent changes of blood flow in neighboring tissue sub-volumes have been observed [37], again indicating great variability (Figure 1B).

**Note**: The preclinical data presented in Figure 1D contradict the notion proposed earlier that the improvement of the tumor oxygenation (*pO*_2_) upon mHT might be due to a decreased O_2_ consumption rate (*V_O_*_2_) [12]. Compared to the 35 °C data, TBF at 40 °C increased by 30%, the O_2_ supply by 26%, the O_2_ consumption *V_O_*_2_ by 47%, the O_2_ extraction rate by 17%, and the tumor tissue *pO*_2_ by 50%. Our own data are indicative of a significantly reduced O_2_ consumption rate of tumors in situ only at temperatures ≥43.5 °C for >30 min (Figure 1B,C). Due to equivalent (“proportionate”) increases in microcirculatory blood flow (red blood cell flux) and in O_2_-consumption rates during mHT, no changes in the mitochondrial redox status were reported [38].

Mechanisms most probably involved in ***transient blood flow improvements*** during mHT (and ≤ 1 h post-heat) are shown in Figure 2. In this flowchart, mHT-induced (patho-)physiological mechanisms contributing to a short-term increase in TBF and, thereby, to a more efficient convective (i.e., blood-borne) transport are summarized.

Short term mechanisms include:Primary dilation of co-opted vessels within tumors;Thermoregulatory dilation of upstream blood vessels in the normal tissue adjacent to the growing tumor, a regulation that leads to secondary flow increases (“re-perfusion”) through downstream tumor vessels in series with the normal tissue vascular bed [39];Distinct reduction of viscous resistance to flow due to significant improvements in the key rheological parameters that determine blood flow behavior [40]. *In vitro*, a temperature rise of 1 K significantly decreases the blood viscosity by ≈ 3.5% and the plasma viscosity by ≈ 2.5% [41,42,43]. The relative kinematic viscosity of blood (at different hematocrit values) and of plasma as a function of temperature is shown in Figure 3 [40]. Taken together, these mHT-induced changes in key parameters clearly improve the blood flow behaviour.

**Note**: Together with the fall in the geometric resistance due to vasodilation observed in mHT, it may be expected that the above-mentioned reduction in viscous resistance may also result in a drop of total resistance to flow (mmHg∙min∙mL^−1^) in tumors *in vivo*. During hyperthermic perfusion of tumors in situ, a rise in tissue temperature from 37 °C up to 39.5 °C caused a drop of total resistance to the flow of 16%, i.e., ≈ 6.5%/K [44]. This result may—at least partly—be explained by a significant loss of red blood cell deformability in severely acidic tumors *in vivo* [45].

Sustained “re-perfusion” up to 24–48 h after heating has been attributed to a strong (≈ 80%) reduction of interstitial fluid pressure (IFP in normal tissues: −2 to 0 mmHg vs. IFP in tumors: up to +10–30 mmHg, “interstitial hypertension“) [31,46,47,48]. Increased blood flow in the latter case is the result of restoring effective perfusion pressures in the tumor microcirculation [49].

The therapeutically relevant increases in TBF upon mHT, which are sustained up to 48 h post-HT, most probably result from a series of (plausible and/or experimentally proven) mechanisms, as shown in Figure 2. Besides (i) mHT-induced lowering of IFP, followed by subsequent restoration of adequate perfusion pressure (see above), another mechanism for sustained blood flow increases for 24–48 h is based on (ii) the mHT-induced activation of ERK-/Akt-pathways [50,51] followed by HIF-1α activation with subsequent increased VEGF-secretion, the latter leading to robust tumor angiogenesis and rise in TBF (see Figure 4).

### 3.2. Enhanced Tumor Oxygenation Status upon Localized Mild Hyperthermia

#### 3.2.1. A Multifactorial, Complex Scenario Is Involved in the Transient Improvement of Tumor Oxygenation upon Mild Hyperthermia

In general, an adequate O_2_ supply in malignant tumors (primaries and metastases) is spatially and temporally restricted by (i) inadequate perfusion (convective transport) and (ii) limited diffusion (diffusive transport), both leading to heterogeneously distributed hypoxia (i.e., pO_2_ < 10 mmHg; severe hypoxia: pO_2_ < 1 mmHg). Perfusion- and diffusion-limited hypoxia is detectable even in early growth stages. A major goal of localized mHT in the combined treatment setting is the improvement of the tumor oxygenation status, which critically impacts inter alia radio-sensitivity, the efficacy of a series of anticancer drugs, protective antitumor immune responses, tumor progression (tumor aggressiveness), and finally patient outcome.

As shown in Figure 4, localized mHT can improve the tumor oxygenation status via several mechanisms:

**Role of oxygen availability:** As outlined above, mHT can lead to transient improvements of TBF. In most normal tissues, increases in nutritive blood flow usually result in concurrent increases in the O_2_ availability (O_2_ supply = blood flow × arterial O_2_ concentration) to a comparable extent, assuming a constant O_2_ concentration in the arterial blood (≈ 20 mL O_2_/dL blood).

In patient tumors, a significant fraction of arterial blood (up to 25–30% [31,52,53]) can be shunted to the venous side without participating in the microvascular exchange processes. This fraction may significantly increase during mHT, thus excluding the assumption of a straightforward proportionality between TBF and O_2_ availability (at constant arterial O_2_ content).

**Note**: During the HT treatment of anemic cancer patients (*cHb* < 12 g/dL), a special situation regarding the O_2_ supply conditions has to be taken into consideration. In this situation, two counteracting mechanisms determining the O_2_ supply of tumors have to be discussed. On the one hand, the O_2_ availability is reduced due to the lower O_2_ concentration in the arterial blood; on the other hand, the oxygen release from oxyhemoglobin (HbO_2_) into the surrounding tissue is enhanced in anemic patients due to increased concentrations of hemoglobin-bound 2,3-bisphosphoglycerate (2,3-BPG), which leads to a right-shift of the HbO_2_ dissociation curve (see below). However, the latter adaptation can only partially compensate for anemic hypoxemia.

**Role of oxygen diffusivity:** On average, in situ tumors have a ≈ 15% higher water content than their tissues of origin (“tumor hyperhydration”) [4]. As a consequence of this *hyperhydration*, heat deposition upon electromagnetic irradiation is higher in tumors than in their corresponding normal tissues. Additionally, hyperhydration causes a distinctly higher O_2_ diffusivity (Fick’s oxygen diffusion coefficient, D_O2_).

*D_O_*_2_ values (O_2_ diffusivities) for tumors are increased by a factor of ≈ 1.9 compared to the tissues of origin [54]. Additional edema formation is often seen upon HT [9], thus increasing hyperhydration and—as a result—further improving O_2_ diffusivity, finally supporting re-oxygenation of the tumor upon mHT.O_2_ diffusivity is also increased by mHT per se. Heating a tissue/tumor to 43 °C increases *D_O_*_2_ by 2.1–4.6%/K [55,56].

**Role of intensified tumor acidosis:** Tissue exposure to mHT triggers a series of events that aggravate tumor tissue acidosis (pH↓), finally reaching extracellular pH values of ≈ 6.20:Besides anaerobic glycolysis (because of tumor hypoxia), cancer cells rely on aerobic glycolysis (due to metabolic reprogramming, a core hallmark of cancer [57]). Both pathways produce high amounts of lactate and protons H^+^ (“lactic acid”), which are exported into the extracellular space, leading to extracellular acidosis (mean *pH_e_* ≈ 6.75). Aerobic glycolysis is stimulated by mHT-induced activation of HIF-1α, leading to an intensified Warburg effect for 24–48 h (schematically shown in Figure 5).In addition, tissue heating intensifies ATP hydrolysis with proton production as well as inhibiting the Na^+^/H^+^ antiport of the cell membrane [10].mHT per se intensifies tissue acidosis due to changes in chemical equilibria of the intra- and extracellular buffer systems: Δ*pH*/Δ*T* = −0.016 pH units/K [10].

Upon mHT, pH_e_ values steadily decrease, finally reaching ≈ 6.20 [58,59,60,61,62,63]. This effect is temporary, i.e., it is observed only during mHT-application and until ≤ 6 h post-heat, leading to parallel effects only in this period.

Intensified acidosis per se leads to a right shift of the HbO_2_-dissociation curve, regardless of the mechanisms causing acidosis (pH-Bohr effect) [63,64,65]. As shown in Figure 6A, the respective difference in the pO_2_ values is ≈ 9 mmHg at 50% HbO_2_ saturation, which is comparable to the right shift when considering the fetal (HbF) and the maternal (HbA) blood at term.

mHT per se also leads to a temperature-dependent right shift of the HbO_2_-dissociation curve (*ΔpO*_2_ ≈ 13 mmHg at 50% HbO_2_ saturation, see Figure 6B) [63,64,65]. Both mechanisms (i.e., intensified acidosis and temperature rise) result in an enhanced O_2_ release from oxyhemoglobin into the surrounding tissue, thus supporting tumor tissue “reoxygenation”.

During mHT, a significant rise (+30%) of the CO_2_ partial pressures in the tumor-venous blood was found, paralleled by the pH drop mentioned above, thus amplifying the right-shift of the HbO_2_-dissociation curve (CO_2_-Bohr effect) [44,63,64,65].

**Role of hyperthermia-activated ERK/Akt-signaling pathways**: In clinical settings, improvements in oxygenation status have occasionally been observed up to 24–48 h post-heating, most probably accompanying longer-lasting increases in TBF upon VEGF-driven forced angiogenesis [50], and—less likely—by HIF-1α-mediated inhibition of mitochondrial functions (see Figure 4). As already mentioned above, changes in the mitochondrial redox status were not detected shortly after finishing mHT [38]. As shown in Figure 1D, the O_2_ extraction steadily increases with temperature, clearly speaking against the notion of impaired OXPHOS upon mHT. This latter finding agrees with the basic results of Gullino et al. [66], who investigated the relationship between temperature and blood supply, and consumption of oxygen in rat mammary carcinomas.

#### 3.2.2. Experimental and Clinical Evidence for Improved Tumor Oxygenation upon Localized Mild Hyperthermia: Updated Data Analysis

Considering relevant data describing the tumor oxygenation status upon clinical mHT, the relationship between mean baseline (pre-HT) tissue *pO*_2_ (NT-pO_2_) and post-HT *pO*_2_ values (PT-pO_2_) can be linearly fitted for experimental and patient tumors upon mHT (see Figure 7A). These data suggest an equivalency of the HT techniques used in preclinical and clinical experiments to achieve this effect. Considering additional measurements of subcutaneous *pO*_2_ values in the clinical setting (red dot [67]), steady state mHT leads to a 1.64-fold increase in the mean *pO*_2_ values. Considering only experimental rodent tumors [6,7,35,61,62,68,69,70,71,72,73,74,75], primary canine [76,77], and human tumors (breast cancers, soft tissue sarcomas [78,79,80] with baseline values between 1 and 10 mmHg (i.e., hypoxic tumors), mHT for 30–60 min almost doubles the pre-heat *pO*_2_ values (see Figure 7B), again irrespective of the heating technique (e.g., hyperthermic perfusion of tumors in situ, microwaves, radiofrequencies, water-filtered Infrared A), even if data of reasonably reliable experiments with water bath immersion at 38.5–41.5 °C were included [75]. In these experiments, the mean pO_2_ before mHT was 7 mmHg (hypoxic fraction HF < 10 mmHg: 67%); during mHT, the average pO_2_ was 17 mmHg (HF: 46%), with a rapid return (within ≈ 10 min) to the pre-HT oxygenation status.

Significant improvements in the tumor oxygenation status described above are fully comparable with data from preclinical experiments upon short-term spontaneous breathing pure oxygen (100% O_2_). In these experiments, steady state hyperoxic *pO*_2_ values of 400–450 mmHg within the arterial blood were reached. In contrast to mHT, an immediate, rapid drop of tumor *pO*_2_ to baseline values was observed upon return to room air breathing (≤ 1–2 min) [81].

Exposure of severely hypoxic tumors (average NT-pO_2_ ≈ 1 mmHg) to mHT for 30–60 min increases the HT-pO_2_ to a mean of ≈ 3 mmHg. As shown in Figure 8, this small improvement in the oxygenation status increases the relative radiosensitivity (oxygen enhancement ratio, OER) from 1.32 to 1.84 (Δ_OER_ = 0.52) due to the relatively steep slope of this section of the OER curve (blue range, A). In hypoxic tumors (e.g., locally advanced or recurrent breast cancers) with mean baseline *pO*_2_ values of ≈ 5 mmHg, mHT can cause a rise up to ≈ 10 mmHg, which is accompanied by an increase in OER from 2.10 to 2.39 (Δ_OER_ = 0.29), caused by a less steep course of the OER curve (green range, B). In less hypoxic tumors (NT-pO_2_ ≈ 17 mmHg), mHT leads to an increment up to a mean HT-pO_2_ of ≈ 32 mmHg, yielding an increase in OER from 2.53 to 2.79 (Δ_OER_ = 0.26; red range, C). From these data, it may be concluded that the mHT-related gain in relative OER is greatest in the very hypoxic tumors.

**Note**: As mentioned above, tumor oxygenation is considered as the key factor to substantially increase radiosensitivity. Therefore, mHT should be applied immediately/as close as possible before radiotherapy (RT). Using this schedule for recurrent breast cancers, re-RT doses could be significantly reduced compared to RT doses reported earlier [82].

## 4. Conclusions and Outlook

Localized mHT can lead to transient (and occasionally sustained) improvements in TBF, both conditions being caused by a variety of different mechanisms. Increased O_2_ levels upon mHT are roughly equivalent to changes in TBF and O_2_ availabilities. However, a series of mechanisms, in addition to the improved TBF, significantly contribute to the increased oxygenation observed upon mHT, such as (i) increased diffusivities and (ii) intensified tumor acidosis and mHT per se, both leading to a right-shift of the HbO_2_ dissociation curve, thus enhancing O_2_ release from HbO_2_, and contingently (iii) HIF1-α-mediated inhibition of mitochondrial functions, a condition sometimes proposed. Whether mHT-induced suppression of cell proliferation and apoptosis of “aerobic” tumor cells (e.g., [50]), eventually reducing oxygen consumption and thus improving the oxygenation status, can be considered valid is the subject of current investigations, especially when translating this observation into clinical settings. Summarizing the result of mHT application, it can be stated that the transient reduction of tumor tissue hypoxia is—most probably—the result of a reduction of both perfusion-limited and diffusion-limited hypoxia.

Localized mHT (39–42 °C for 30–60 min) leads to an almost doubling of initial O_2_ levels in tumors, independent of the heating technique used. In essence, we recommend considering the provided data in treatment protocols comprising mHT, particularly when sequences and time intervals in combined treatment modalities (e.g., thermo-radio-immuno-oncology) are discussed.

## Figures and Tables

**Figure 1 cancers-15-01394-f001:**
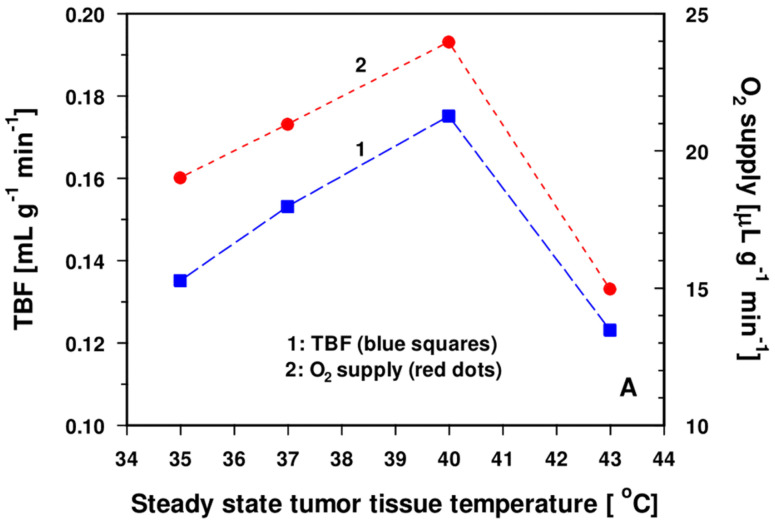
Changes of relevant physiological parameters upon localized mHT as a function of tumor temperature. Results show tumor blood flow (TBF) and oxygen (O_2_) supply (=TBF × arterial O_2_ concentration, panel (**A**)), microvascular oxyhemoglobin (HbO_2_) saturation, and tumor tissue O_2_ partial pressure (*pO*_2_, panel (**B**))*,* in situ O_2_ consumption (*V_O_*_2_) and—completely independent of blood flow—the O_2_ consumption rate of isolated tumor cells *in vitro* (*MRO*_2_), panel (**C**)). There is clear evidence that TBF and the directly flow-dependent parameters have a maximum between 39.5 °C and 40.5 °C, whereas—at maintained, unrestricted O_2_ supply under *in vitro* conditions—*MRO*_2_ values have a distinct maximum at 42 °C with a decline at temperatures ≥ 43 °C (at maintained, adequate O_2_ supply under *in vitro* conditions). The oxygen extraction rate (= O_2_ uptake/O_2_ supply), a measure of the adequacy of the O_2_ supply to the tissue, increases with rising temperature (panel (**D**)). Oxygen diffusivity (*D_O_*_2_*)* increases linearly with temperature (panel (**E**)). Based on data provided in [32,33,34,35]. Values are averaged means.

**Figure 2 cancers-15-01394-f002:**
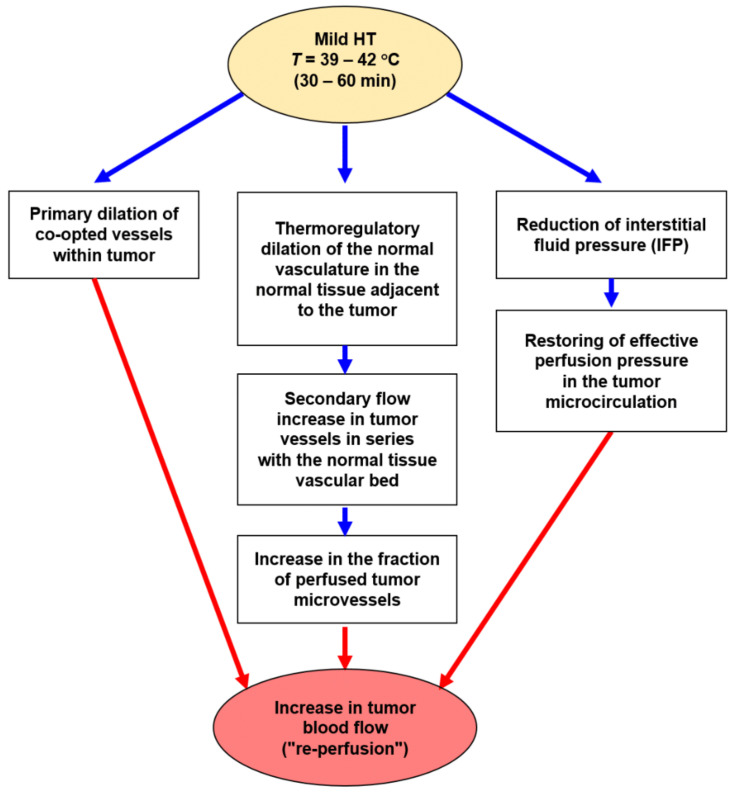
Mild hyperthermia leads to transient increases in tumor blood flow through primary dilation of co-opted pre-existing blood vessels (left pathway) and secondary flow increase (“re-perfusion”) through tumor blood vessels following thermoregulatory “upstream-vasodilation” in the normal tissue adjacent to the “downstream” tumor vessels (central pathway). More sustained improvements may be due to “resumptions” of blood flow due to drastic reduction in interstitial hypertension (i.e., lowering of interstitial fluid pressure, right pathway) and activation of molecular pathways that stimulate VEGF-secretion and thus forced tumor angiogenesis (see Figure 4, left pathway).

**Figure 3 cancers-15-01394-f003:**
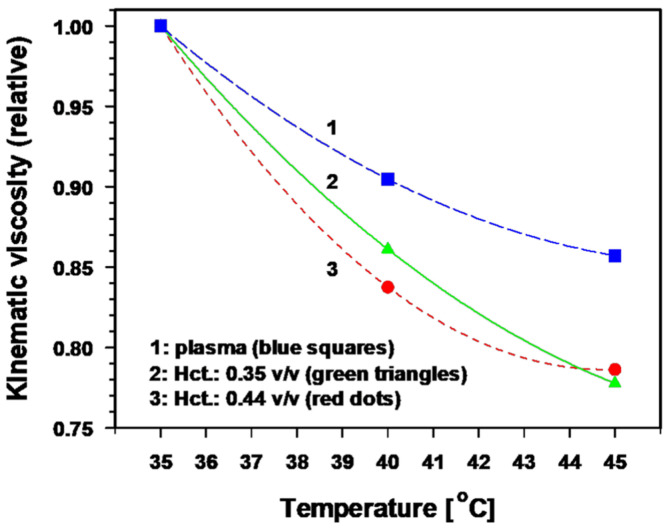
Relative kinematic blood viscosity and plasma viscosity as a function of temperature at physiological pH (based on data provided in [45]). Additional information: At 37 °C, plasma viscosity is ≈ 1.3 cPoise (cP), and apparent viscosity of whole blood is ≈ 3.5 cP (shear rate: 150 s^−1^).

**Figure 4 cancers-15-01394-f004:**
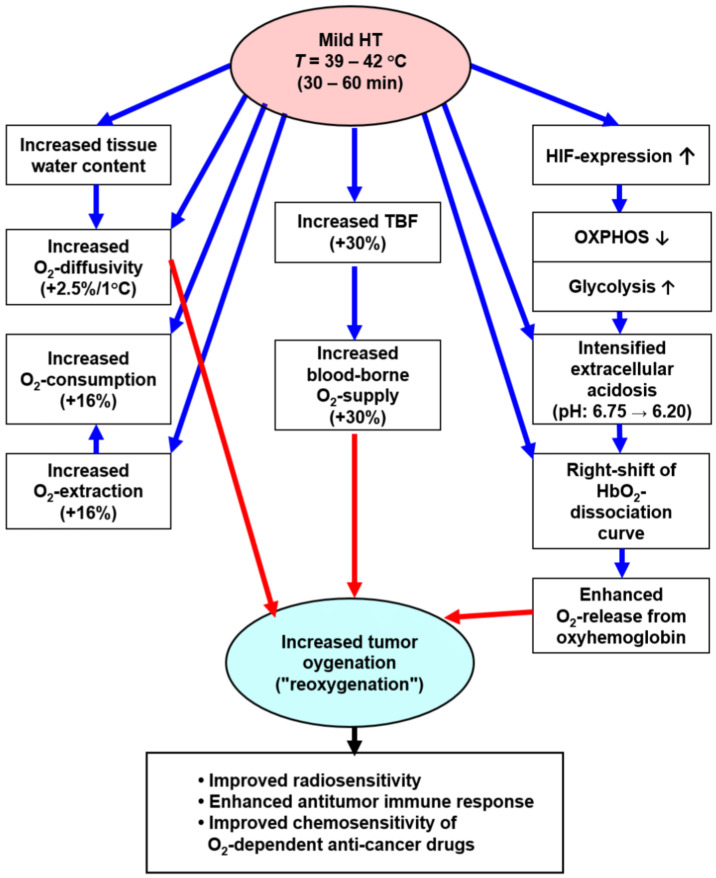
Mechanisms responsible for the improvement of the O_2_ status upon localized mHT through enhancement of the O_2_ availability (central pathway), increased O_2_ diffusivity due to hyperhydration of tumor tissues and direct mHT-impact (left pathway), and intensified tumor acidosis leading to a right-shift of the HbO_2_-dissociation curve, thus facilitating the O_2_ release from the blood into the tissue (right pathway). Most probably, the mHT-induced increment in the O_2_ consumption is compensated by a comparable increase in the O_2_-extraction (left pathway).

**Figure 5 cancers-15-01394-f005:**
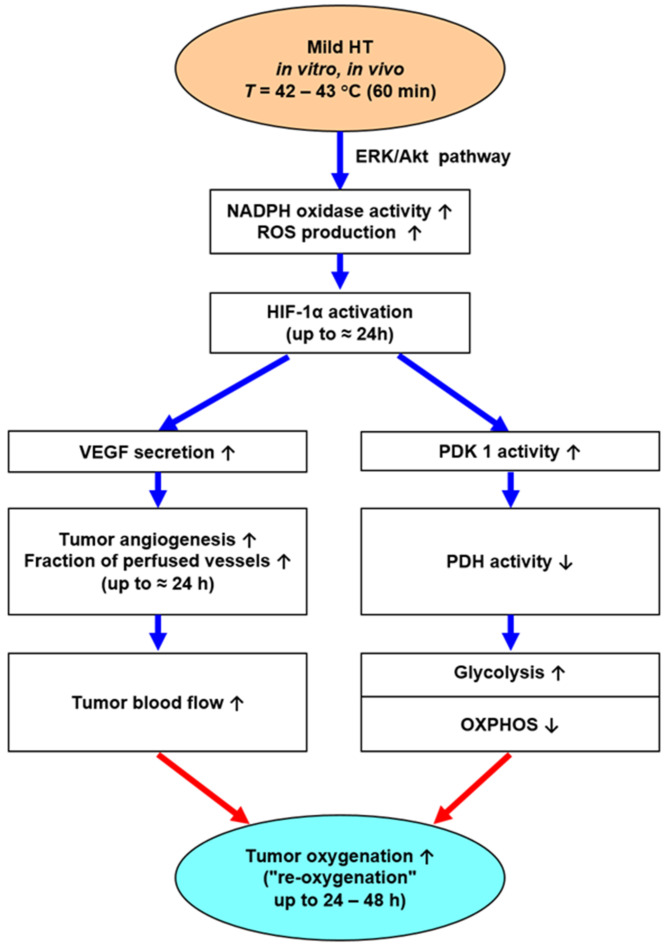
Heat-sensitive ERK/Akt-pathways (central pathway), metabolic reprogramming (right pathway), and VEGF-triggered robust angiogenesis (left pathway) and their putative role in sustained (up to 24–48 h) improvement of the tumor oxygenation status.

**Figure 6 cancers-15-01394-f006:**
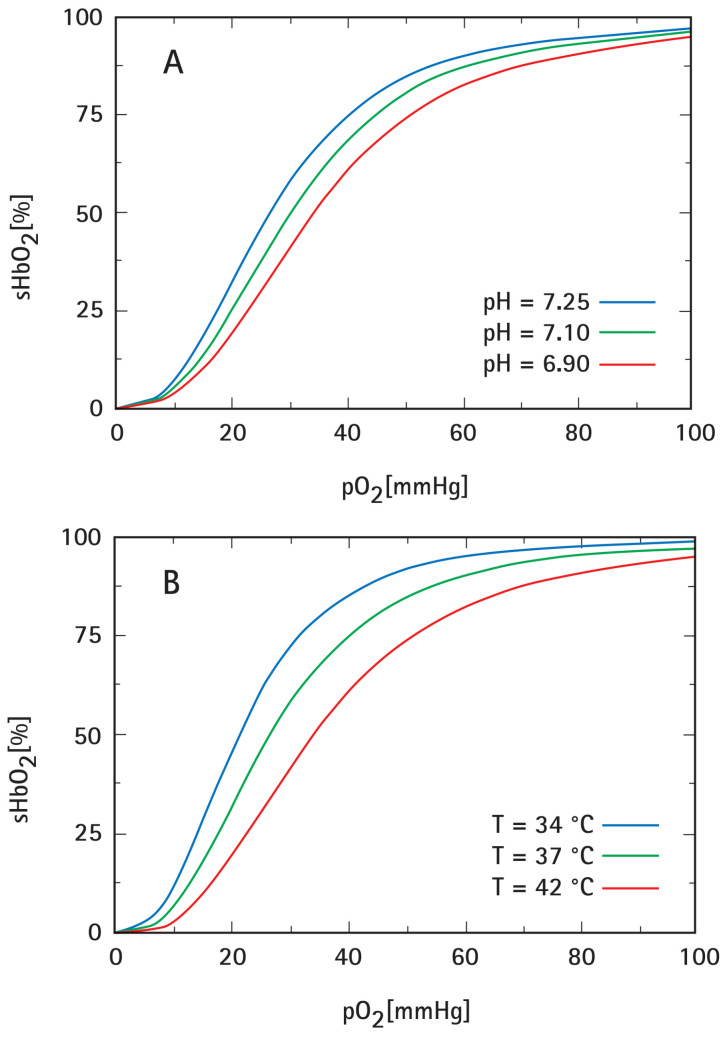
Right-shift of the HbO_2_-dissociation curve resulting from intensified tumor acidosis (**A**) and hyperthermic tumor tissue temperatures (**B**), both facilitating the O_2_ release from oxyhemoglobin (HbO_2_), i.e., enhancing O_2_ unloading from red blood cells into the surrounding tissue.

**Figure 7 cancers-15-01394-f007:**
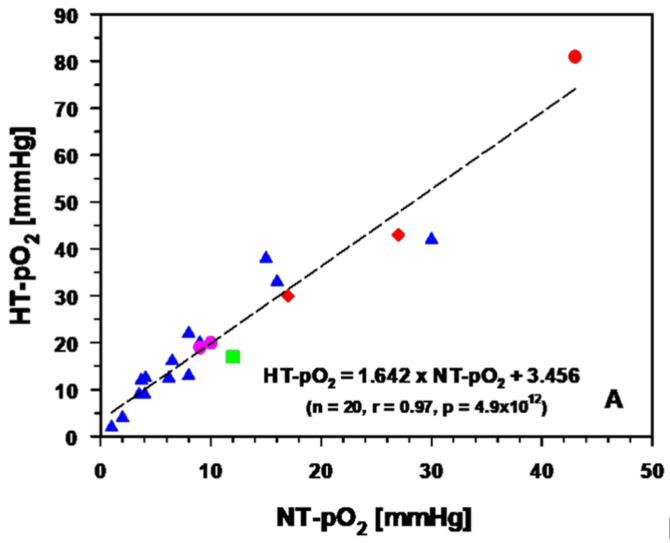
Linear relationship between mean baseline (pre-HT) tumor tissue pO_2_ (NT-pO_2_) and mean post-HT pO_2_ values (HT-pO_2_) in experimental and patient tumors using various heating techniques. In (**A**), the relationship includes normal tissue (dermis) measurements upon heating with water-filtered infrared-A irradiation (wIRA), red dot [67]). Data clearly demonstrate the equivalence of the various heating techniques chosen (blue triangles: microwave heating of rodent tumors; purple dots: microwave heating of patient tumors; red diamonds: wIRA-induced heating of experimental tumors; green square: mean pO_2_ value of all data presented). In general, the use of mHT for 30–60 min leads to a 1.6-fold increase in mean pO_2_. In (**B**) only data of hypoxic tumors (mean pO_2_ < 10 mmHg) are shown. Here, post-mHT pO_2_ values almost doubled.

**Figure 8 cancers-15-01394-f008:**
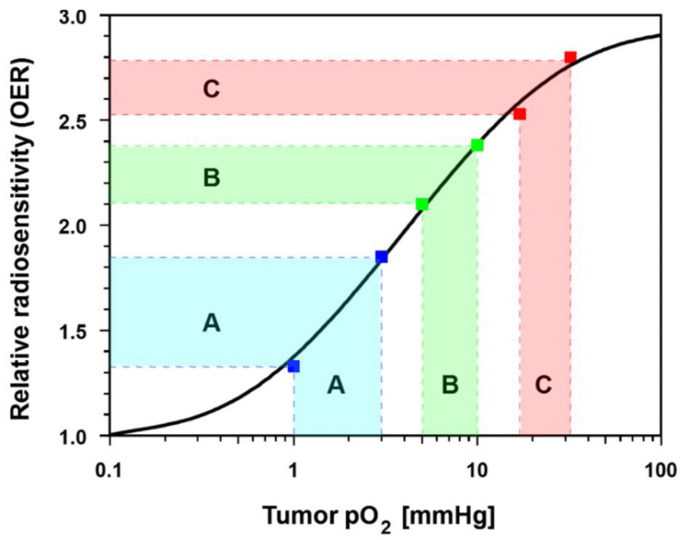
In severely hypoxic tumors (mean pretreatment *pO*_2_ ≈ 1 mmHg) exposed to mHT for 30–60 min, the mean post-HT *pO*_2_ was ≈ 3 mmHg, resulting in a rise in the Oxygen Enhancement Ratio (OER, relative radiosensitivity) from 1.32 to 1.84 (blue range, A). In hypoxic tumors (mean pre-HT *pO*_2_ = 5 mmHg), HT caused a rise to 10 mmHg, which was accompanied by an increase in OER from 2.10 to 2.39 (green range, B). In less hypoxic tumors (mean pre-HT *pO*_2_ = 17 mmHg), mHT led to a *pO*_2_ increment up to 32 mmHg, resulting in an increase in OER from 2.53 to 2.79 (red range, C).

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
