# Peer review of "From Localized Mild Hyperthermia to Improved Tumor Oxygenation: Physiological Mechanisms Critically Involved in Oncologic Thermo-Radio-Immunotherapy"

_cancers, 2023, doi:10.3390/cancers15051394_

Round 1

Reviewer 1 Report

The manuscript of Vaupel et al. is aimed at summarizing the current knowledge concerning the links between hyperthermic treatments and tumor oxygenation, and the mechanisms of their beneficial effects in oncologic therapies. In specific, Authors review data derived from a systematic literature review of this topic. In addition, Authors also provide summary figures using data from their own, already published manuscripts describing their findings from diverse preclinical studies. Based upon the reviewed literature and their own data, Authors conclude that tumor oxygenation enhancement involves a diverse set of signaling pathways and cannot be fully attributed to an enhancement of tumor blood flow induced by mild hyperthermic treatment. The review is supported by 8 Figures and cites 82 references.  

The manuscript fits the scope of the Special Issue: Advances in Radiation Immuno-Oncology: Progress at the Interface of Radiation Oncology and Immunotherapy of “Cancers”, and the manuscript is of interest for the readers of the journal. The manuscript is well written and provides a balanced review of the literature.

This reviewer notes only some minor issues:

1.     Page 3. Some references (20 and 21) are in different size than the text

2.     Page 5, Line 218, refering to Figure 1: Authors should specifically list their own publications that were used for constructing Figure 1 panels. 

3.     Figures 1and 3: the Figure legends should be simplified (e.g. like in Figure 6).

4.     Figure 1: Authors should try to align the 5 panels in two rows for better visual presentation

Author Response

The manuscript of Vaupel et al. is aimed at summarizing the current knowledge concerning the links between hyperthermic treatments and tumor oxygenation, and the mechanisms of their beneficial effects in oncologic therapies. In specific, Authors review data derived from a systematic literature review of this topic. In addition, Authors also provide summary figures using data from their own, already published manuscripts describing their findings from diverse preclinical studies. Based upon the reviewed literature and their own data, Authors conclude that tumor oxygenation enhancement involves a diverse set of signaling pathways and cannot be fully attributed to an enhancement of tumor blood flow induced by mild hyperthermic treatment. The review is supported by 8 Figures and cites 82 references.  

The manuscript fits the scope of the Special Issue: Advances in Radiation Immuno-Oncology: Progress at the Interface of Radiation Oncology and Immunotherapy of “Cancers”, and the manuscript is of interest for the readers of the journal. The manuscript is well written and provides a balanced review of the literature.

This reviewer notes only some minor issues:

  1. Page 3. Some references (20 and 21) are in different size than the text

Size was adapted

  1. Page 5, Line 218, refering to Figure 1: Authors should specifically list their own publications that were used for constructing Figure 1 panels. 

Line 218 .. since 1980 [32-35], using

  1. Figures 1and 3: the Figure legends should be simplified (e.g. like in Figure 6).

Legends of Figs 1 and 3 cannot be simplified without loosing relevant information.

  1. Figure 1: Authors should try to align the 5 panels in two rows for better visual presentation

The 5 panels may be aligned by the publisher.

Reviewer 2 Report

I am grateful for the opportunity to review manuscript ID: cancers-2155617, entitled “From localized mild hyperthermia to improved tumor oxygenation: Physiological mechanisms critically involved in oncologic thermo-radio-immunotherapy.”

The authors present an excellent review of different pathways of the impact of mild hyperthermia on the tumor micromilieu, from blood flow to oxygen bioavailability. Modulation strategies of the tumor microenvironment are in the focus of interest in the era of novel immunotherapies.

The manuscript is clear and firm. Citations are relevant and up-to-date. Data presentation is appropriate.  Conclusions are consistent with the presented data. The work fit well the scope of Cancers. The conclusions seem to be interesting to the readership of the journal. In general, the manuscript deserves publication.

Detailed comments:

Please consider reporting the results of the search and selection process, from the number of records identified in the search to the number of studies included in the review, ideally using a flow diagram (following PRISMA recommendations).

Please check minor formatting issues (e.g. citation format in lines 109 and 120; use of brackets in line 449, and omitting ORCIDs from author contribution in lines 501-504).

Author Response

I am grateful for the opportunity to review manuscript ID: cancers-2155617, entitled “From localized mild hyperthermia to improved tumor oxygenation: Physiological mechanisms critically involved in oncologic thermo-radio-immunotherapy.”

The authors present an excellent review of different pathways of the impact of mild hyperthermia on the tumor micromilieu, from blood flow to oxygen bioavailability. Modulation strategies of the tumor microenvironment are in the focus of interest in the era of novel immunotherapies.

The manuscript is clear and firm. Citations are relevant and up-to-date. Data presentation is appropriate.  Conclusions are consistent with the presented data. The work fit well the scope of Cancers. The conclusions seem to be interesting to the readership of the journal. In general, the manuscript deserves publication.

Detailed comments:

Please consider reporting the results of the search and selection process, from the number of records identified in the search to the number of studies included in the review, ideally using a flow diagram (following PRISMA recommendations).

According to our correspondence with Ms. Kiran Yu, Assistant Editor, MDPI, on December 28, 2022, this is a “basic” review paper. Only “Systematic reviews” need to show PRIMA diagrams.

Please check minor formatting issues (e.g. citation format in lines 109 and 120; use of brackets in line 449, and omitting ORCIDs from author contribution in lines 501-504).

Citation format in lines 109 and 120 have been corrected. ORCIDS (shown in lines 501-504) may be replaced by the Publisher.

The authors want to thank both reviewers for constructive suggestions.
